# Detection and Phylogenetic Characterization of a Novel Adenovirus Found in Lesser Mouse-Eared Bat (*Myotis blythii*) in South Kazakhstan

**DOI:** 10.3390/v15051139

**Published:** 2023-05-10

**Authors:** Kobey Karamendin, Aidyn Kydyrmanov, Temirlan Sabyrzhan, Sardor Nuralibekov, Yermukhammet Kasymbekov, Yelizaveta Khan

**Affiliations:** Scientific Production Center of Microbiology and Virology, 105 Bogenbay Batyr Str., Almaty A25K1G0, Kazakhstan; kydyrmanov@yandex.kz (A.K.); temirlans2019@gmail.com (T.S.); nuralibekovs@mail.ru (S.N.); kasymbek.ermuxan@mail.ru (Y.K.); lizaveta4ka@list.ru (Y.K.)

**Keywords:** bat, mastadenovirus, *Myotis blythii*, transmission, Kazakhstan

## Abstract

Bats are an important natural reservoir of various pathogenic microorganisms, and regular monitoring is necessary to track the situation of zoonotic infections. When examining samples from bats in South Kazakhstan, nucleotide sequences of putative novel bat adenovirus (AdV) species were found. Estimates of amino acid identities of the hexon protein have shown that potentially novel Bat mastadenovirus BatAdV-KZ01 shared higher similarity with monkey Rhesus adenovirus 59 (74.29%) than with Bat AdVs E and H (74.00%). Phylogenetically, BatAdV-KZ01 formed a separate clade, distant from Bat AdVs and other mammalian AdVs. Since adenoviruses are essential pathogens for many mammals, including humans and bats, this finding is of interest from both scientific and epidemiological points of view.

## 1. Introduction

It was estimated that 75% of all emerging human infections originate in other animals, and bats are an important reservoir of many zoonotic viruses [1]. More than 1400 bat species are known worldwide, constituting about one-fifth of all mammal species [2]. Fifteen thousand nine hundred nine bat-associated viruses were registered in the ZOVER database [3], and 19 species were recognized as pathogenic for humans, including Ebola, Nipah, and Hendra viruses [4]. The last pandemics caused by severe acute respiratory syndrome coronaviruses presumably originated from the bat reservoir [5]. It was defined that bat virome varies depending on the geographical location and the bat species [6]. About 30 species of bats are registered in Kazakhstan, which is nearly half of all species of the Palearctic [7]. The wealth of bat fauna is due to Kazakhstan’s vast and diverse climatic zones.

Adenoviruses are double-stranded DNA viruses that infect almost every major vertebrate class [8]. Adenoviruses cause many diseases, mainly in the respiratory and gastrointestinal systems, but some of them may be asymptomatic [9]. Today, ten adenovirus species are registered in bats: Bat mastadenoviruses from A to J [10]. The role of adenoviruses in regulating bats’ population size and disease prevalence has been poorly studied. The identification of adenovirus from a bat in this study is the first for Kazakhstan and the Central Asian region. Gull adenovirus [11] and Cormorant adenovirus (unpublished data) found in the wild fauna of Kazakhstan are known to have caused mass mortality among birds of the corresponding species. Intensive studies of bat viruses are carried out in the Central Asian region, and Bat alphacoronavirus and Lyssavirus (rabies) have been identified [12]. This study reported the identification of a potentially novel bat adenovirus provisionally named BatAdV-KZ01 different from those ten known Bat mastadenovirus species.

## 2. Materials and Methods

A retrospective analysis of the viral metagenome of three bats (*Myotis blythii*) accidentally caught in ornithological nets during bird ringing in south Kazakhstan (43.77° N 69.50° E) in 2008 was conducted. Rectal swabs were taken and placed in tubes with a viral transport medium. Swab samples were centrifuged at 3200 RPM for 15 min (Eppendorf 5417R, Rotor FA-45-24-11, Hamburg, Germany) and filtered through a 0.45 μm filter (Membrane Solutions, Auburn, WA, USA). Samples were treated with a mix of nucleases: Benzonase (Sigma-Aldrich, St. Louis, MO, USA), Turbo DNAse, DNAse I, RNAse A, and RNAse T1 (Thermo Fisher, Vilnius, Lithuania) to eliminate host RNA and DNA. Viral nucleic acids were extracted using the QIAamp Viral RNA Mini Kit (Qiagen, Hilden, Germany) according to the manufacturer’s instructions.

For massive parallel sequencing, libraries were constructed from each sample separately using the NEBNext Ultra DNA Library Preparation kit (New England Biolabs, Ipswich, MA, USA). The size and quality of libraries were verified on a Bioanalyzer 2100 instrument (Agilent Technologies, Waldbronn, Germany). MiSeq Reagent v.3 kit was used for sequencing on a MiSeq sequencer (Illumina, St. Louis, MO, USA). The obtained data were trimmed and, de novo, assembled using Geneious Prime 2022 software (Biomatters, Auckland, New Zealand). Then the assembled contigs were subjected to BLASTx search in the local viral reference database as described in Diamond software v.2.1.6 [13]. BLAST hits with lengths of more than 200 nucleotides (nt) were considered significant at E value < 1 × 10^−5^, and the potential viral sequences were subjected to a new BLASTx search in the NCBI non-redundant protein database.

Estimates of amino acid identities between BatAdV-KZ01 and other mastadenoviruses were calculated in Geneious Prime 2022: alignment of sequences in Clustal Omega 1.2.2 inferred the percent identity matrix.

Phylogenetic analysis was performed by applying a Maximum Likelihood method and Le_Gascuel_2008 model [14] that was identified as the most appropriate sequence evolution model on the basis of Akaike Information Criterion (AIC) and Bayesian Information Criterion (BIC) ranking via ModelTest, as implemented in Mega 11 software [15]. The Genbank accession number for the hexon, IVa2, and pVIII genes partial nucleotide sequences analyzed in this study are OQ599371, OQ599372, and OQ599373, respectively.

## 3. Results

AdV sequences were found in one of three sequenced samples from *M. blythii* bats. This positive sample contained 162,092 raw sequencing reads, and the de novo assembly produced 10,689 contigs, and three of them represented adenovirus fragments of the hexon (219 nt), pVIII (242 nt), and IVa2 (158 nt) genes. Estimation of putative amino acid sequence identities of these three genes with those from other Bat mastadenoviruses A–J was conducted (Table 1). Hexon and IVa2 were the most conserved, while pVIII showed high variability. Calculating the amino acid sequence identities of the hexon gene showed that BatAdV-KZ01 shared a maximum identity of 70.00% with Bat mastadenoviruses E and H. By IVa2 gene, it was close to Bat mastadenoviruses E and G (74.00%). However, BatAdV-KZ01 was closer to the monkey Rhesus adenovirus 59 (74.29%) by hexon gene and to Human adenovirus 40 (78.00%) and Polar bear mastadenovirus 1 (80.00%) by IVa2 gene. By pVIII gene, it was closer to Bat mastadenovirus F and Harbour porpoise adenovirus 1 (59.26% and 59.21%, respectively), but this gene is highly variable, as we noted earlier.

Phylogenetic analyzes showed that BatAdV-KZ01 formed a separate cluster by hexon gene (Figure 1A), significantly distant from both bat and other mammalian adenoviruses. At the same time, by the hexon gene, other Bat mastadenoviruses were divided into three phylogenetic groups. Bat mastadenoviruses D, E, F, H, and I formed a separate cluster of only bat adenoviruses. Bat mastadenoviruses A, B, G, and J formed another cluster, phylogenetically close to Canine mastadenovirus A.

**Figure 1 viruses-15-01139-f001:**
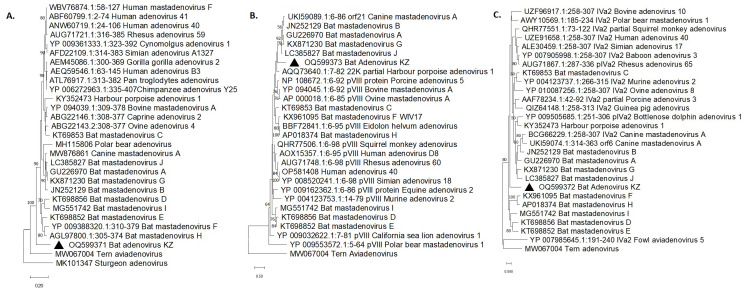
Phylogenetic trees for the partial amino acid (**A**) hexon and (**B**) pVIII and (**C**) IVa2 proteins of the BatAdV-KZ01 adenovirus (indicated with a solid triangle) and other selected species of Mastadenovirus genus. Bootstrap values (1000 replicates) > 35 are indicated at the branch nodes.

**Table 1 viruses-15-01139-t001:** Estimates of Amino Acid Identities between BatAdV-KZ01 and other Bat mastadenoviruses.

Mastadenovirus Species	GenBank Accession №	Genes		
		Hexon	pVIII	IVa2
Rhesus adenovirus 59	MF198453	74.29	50.57	74
Polar bear mastadenovirus 1	MH115806	64.29	41.67	80
Human adenovirus 40	OP581408	62.65	51.72	78
Harbor porpoise adenovirus 1	KY352473	67.14	59.21	68
Bat mastadenovirus A	GU226970	68.57	55.26	70
Bat mastadenovirus B	JN252129	67.14	48.28	64
Bat mastadenovirus C	KT69853	67.14	54.32	72
Bat mastadenovirus D	KT698856	64.29	48.28	72
Bat mastadenovirus E	KT698852	70	47.13	74
Bat mastadenovirus F	KX961095	68.57	59.26	70
Bat mastadenovirus G	KX871230	68.57	49.43	74
Bat mastadenovirus H	AP018374	70	56.32	72
Bat mastadenovirus I	MG551742	68.57	44.74	72
Bat mastadenovirus J	LC385827	67.14	48.28	70

Phylogenetically, Bat mastadenovirus C was the most distant from these two groups. Notably, the obtained phylogenetic data on the hexon gene are entirely consistent with the tree’s topology based on the DNA-dependent DNA polymerase gene when describing novel adenoviruses in ICTV [16].

We also analyzed the highly variable pVIII gene of BatAdV-KZ01 for comparison (Figure 1B). By this gene, BatAdV-KZ01 was closer to the group of viruses consisting of Bat mastadenoviruses A, B, G and J that are, in turn, close to Canine mastadenoviruses A. Bat mastadenoviruses D, E and I, by this gene, were phylogenetically distant from the group consisting of Bat mastadenoviruses C, F and H, which formed a separate cluster. By IVa2 protein (Figure 1C), the topology of the phylogenetic tree was similar to that of the hexon gene, and the BatAdV-KZ01 adenovirus also formed a separate cluster.

Based on one of the species demarcation criteria from the International Committee on Taxonomy of Viruses (ICTV), a phylogenetic distance of >15% (DNA polymerase amino acid sequence) allows separation into different species [17]. BatAdV-KZ01 by hexon gene amino acid sequence was >15% different from known mastadenoviruses and can be assigned to putative a new species. This finding contributed to the knowledge about the bat adenovirus diversity.

## 4. Discussion

Due to increased attention to bat viruses and intensive studies of their virome using deep sequencing, several bat adenoviruses have been identified. The first report was made in 2008 when a strain of Bat adenovirus FBV1 from a Ryukyu flying fox (*Pteropus dasymallus yayeyamae*) was isolated in Japan [18], which was subsequently classified as Bat mastadenovirus A [19]. Since then, nine new Bat adenoviruses from B to J have been identified [20,21,22,23,24,25,26].

Bat AdVs are little studied, and the accumulated data is still insufficient to answer all questions. It was supposed that adenoviruses are strictly specific to the host species, but recent data indicate the ambiguity of this assumption [6]. It was previously shown that Bat AdVs are divided into three groups: group 1, which included A, B, G, and J species, isolated from the Vespertilionidae family and genetically close to canine adenoviruses; group 2, consisting of the only Bat AdV C species from the Rhinolophidae family bats; and group 3, comprised of representatives of species D, E, F, H, and I from Miniopteridae and Pteropodidae bats [23,27]. By the variable pVIII gene, BatAdV-KZ01 clustered into group 1, which includes representatives of the family Vespertilionidae, to which *M. blythii* bat species belong. Based on the hexon gene phylogeny, BatAdV-KZ01 appears to be significantly distant from all mammalian adenoviruses, including bats. BatAdV-KZ01 shared higher similarity (74.29%) with monkey Rhesus adenovirus 59 than with Bat AdVs E and H (74.00%). This phenomenon can be explained by the large diversity of adenoviruses in bats that have a different phylogenetical origin. The striking similarity between the genomic organization of canine and bat adenoviruses was described [28]. Phylogenetic reconstructions also revealed a close relationship between Equine AdV and bat AdV strain TJM [29].

Phylogenetically, BatAdV-KZ01 looks like a predecessor to all known Mastadenoviruses by hexon gene sequence, but further research, including its complete genome study, is required to confirm this assumption.

Additionally, there is still little data on the possibility of cross-species transmission from bats to other mammalians, including humans. The probability of such transmission is supposed by the fact that Bat AdV can grow on human, simian, and canine cell lines [19]. It is known that RNA viruses can switch to other hosts much faster than DNA viruses [30], to which adenoviruses belong. However, DNA virus host shifts occur, and an example is canine mastadenovirus, which was adapted from bats to dogs in the past [28]. It is also known that the successful overcoming of the interspecies barrier depends on the phylogenetic similarity of the hosts [30]. *M. blythii* is a typical member of the Vespertilionidae family, and its cross-species transmission potential is possibly closely related to the potential of other bat species in this family.

It was previously shown that roosting behavior correlates with AdV prevalence: highly social bats are infected by more AdV species [6]. Virus transmission modeling suggested that lyssavirus transmission in *M. blythii* is also density-dependent [31]. *M. blythii*, in which BatAdV-KZ01 was found, forms large colonies from hundreds to several thousand individuals [7]. Therefore, discovering a new adenovirus in one of the three examined samples cannot be considered an exceptional case. It was hypothesized that there are hundreds of unknown and undiscovered AdV species in bats [32]. Further surveillance of bat adenoviruses is essential to understand better their ecology, epidemiology, evolution, and influence on bat populations. It will also help to evaluate the potential risks of cross-species transmissions of adenoviruses from bats.

Limitations of the study: Unfortunately, due to the long time elapsed, isolating the virus in cell cultures was impossible. The complete genome was unavailable, which may lower the resolution of the phylogenetic tree. Additionally, we cannot definitely say that all three gene fragments belong to one virus, and further studies are necessary to confirm this. Anyway, research in the region continues with the hope of re-discovering the virus and completing genome characterization. The polymerase gene sequence was unavailable, so the hexon gene was used for genetic analysis. Both of them are often equally used in characterization of Bat adenoviruses [23,26,27,28].

## Data Availability

Not applicable.

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
