# Peer review of "Detection and Phylogenetic Characterization of a Novel Adenovirus Found in Lesser Mouse-Eared Bat (Myotis blythii) in South Kazakhstan"

_viruses, 2023, doi:10.3390/v15051139_

Round 1
Reviewer 1 Report (Previous Reviewer 3)
The authors answered all points I raised and modified the manuscript accordingly. I consider this version suitable for publication.
Author Response
The authors answered all points I raised and modified the manuscript accordingly. I consider this version suitable for publication.
- Thank you for your comments.
Reviewer 2 Report (Previous Reviewer 2)
The authors have largely addressed my concerns, but I still believe that the study is too thin and that the phylogenetic analysis should be improved. Fig 1A: Bat AdV KZ is still an outgroup as compared to other strains, and most of the phylogenetic nodes aren't well supported by the bootstrap test, i.e., a lot of bootstrap values are less than 80.
Author Response
The authors have largely addressed my concerns, but I still believe that the study is too thin and that the phylogenetic analysis should be improved. Fig 1A: Bat AdV KZ is still an outgroup as compared to other strains, and most of the phylogenetic nodes aren't well supported by the bootstrap test, i.e., a lot of bootstrap values are less than 80.
- Yes, we have added an additional outgroup (the most divergent fish adenovirus) and made everything possible to improve the tree.
Reviewer 3 Report (Previous Reviewer 1)
Minor suggestions:
1. In abstract, line 16: pls revise “..shared higher similarity (74.29%) with monkey Rhesus adenovirus 59 than with Bat AdVs E ..” to “..shared higher similarity with monkey Rhesus adenovirus 59 (74.29%) than with Bat AdVs E ..”.
2. Line 55: pls change ‘0.45 um’ to ‘0.45 μm’.
3. Line 69: revise ‘E value < 10e-5’ to ‘e-value < 1e-5.’.
4. Line 81: suggest to revise “..Le_Gascuel_2008 model [14] identified as the..” to “..Le_Gascuel_2008 model [14] that was identified as the..”.
5. Line 88: change “Myotis blythii” to “M. blythii”, owing to that you have mentioned the name at the begaining of Materials and Methods part. Pls check up throughout the manuscript.
6. Line 90: change “..and their de novo ..” to “..and the de novo ..”.
7. Line 94-95: move the words “and their amino acid identities were estimated using Geneious Prime software” to Materials and Methods part.
8. In Discussion: bat family name should not be in italics. e.g., change “Vespertilionidae” to “Vespertilionidae”, and check up throughout the manuscript.
Author Response
- In abstract, line 16: pls revise “..shared higher similarity (74.29%) with monkey Rhesus adenovirus 59 than with Bat AdVs E ..” to “..shared higher similarity with monkey Rhesus adenovirus 59 (74.29%) than with Bat AdVs E ..”.
- Thank you, corrected.
- Line 55: pls change ‘0.45 um’ to ‘0.45 μm’.
- Corrected
- Line 69: revise ‘E value < 10e-5’ to ‘e-value < 1e-5.’.
- Corrected
- Line 81: suggest to revise “..Le_Gascuel_2008 model [14] identified as the..” to “..Le_Gascuel_2008 model [14] that was identified as the..”.
- Corrected
- Line 88: change “Myotis blythii” to “M. blythii”, owing to that you have mentioned the name at the begaining of Materials and Methods part. Pls check up throughout the manuscript.
- Corrected throughout the manuscript.
- Line 90: change “..and their de novo ..” to “..and the de novo ..”.
- Corrected
- Line 94-95: move the words “and their amino acid identities were estimated using Geneious Prime software” to Materials and Methods part.
- Corrected
- In Discussion: bat family name should not be in italics. e.g., change “Vespertilionidae” to “Vespertilionidae”, and check up throughout the manuscript.
- Thank you, corrected.
Reviewer 4 Report (New Reviewer)
Comments to Authors
This study showed that estimates of amino acid identities of the hexon protein have shown that potentially novel Bat mastadenovirus BatAdV-KZ01 shared higher similarity (74.29%) with monkey Rhesus adenovirus 59 than with Bat AdVs E and H (74.00%).
Authors are kindly requested to emphasize the current concepts about these issues in the context of recent knowledge and the available literature. This articles should be quoted in the References list.
References
1. New geographical and host records of bat fleas (Siphonaptera: Ischnopsyllidae) in Russia. Ann Parasitol. 2022; 68 (1): 121-128. doi:10.17420/ap6801.416.
2. Transmission dynamics of lyssavirus in Myotis myotis: mechanistic modelling study based on longitudinal seroprevalence data. Proc Biol Sci. 2023;290(1997):20230183. doi:10.1098/rspb.2023.0183.
Minor editing of English language required
Author Response
This study showed that estimates of amino acid identities of the hexon protein have shown that potentially novel Bat mastadenovirus BatAdV-KZ01 shared higher similarity (74.29%) with monkey Rhesus adenovirus 59 than with Bat AdVs E and H (74.00%). Authors are kindly requested to emphasize the current concepts about these issues in the context of recent knowledge and the available literature.
These articles should be quoted in the References list. References.
- New geographical and host records of bat fleas (Siphonaptera: Ischnopsyllidae) in Russia. Ann Parasitol. 2022; 68 (1): 121-128. doi:10.17420/ap6801.416.
- Transmission dynamics of lyssavirus in Myotis myotis: mechanistic modelling study based on longitudinal seroprevalence data. Proc Biol Sci. 2023;290(1997):20230183. doi:10.1098/rspb.2023.0183.
- Thank you. The available information was added to the Discussion. The reference for the second article was added. Unfortunately, the full text of the first article was unavailable in PubMed.
Round 2
Reviewer 2 Report (Previous Reviewer 2)
None
This manuscript is a resubmission of an earlier submission. The following is a list of the peer review reports and author responses from that submission.
Round 1
Reviewer 1 Report
The study reported a potentially novel adenovirus found in lesser mouse-eared bat in Kazakhstan by viral metagenome study. Variations of sequences of this viruse (e.g. partial seqences of Hexon, IVa2, and pVIII) implicated the risk of spillover from bats to mammals. But present evidences are insufficient to support the novelity and danger of the viruse. Anyhow, the finding has significance, and it’s still worthy to expect continuous study (e.g. genome level) of the viruse.
1. information of the location capturing the 3 bats is missing. I suggest to add a map to illustrate, or alternatively add the longitude and latitude information in proper position of the manuscript.
2. in line 30: please add references to support the statement.
3. in the Materials and Methods part: line 51-53, pls explain the purpose of the operation: “samples were treated with a mix of nucleases: Ben-51 zonase (Sigma-Aldrich, USA), Turbo DNAse, DNAse I, RNAse A, and RNAse T1 (Ther-52 moFisher, Lithuania).” It’s to eliminate environmental DNA/RNA pollutions? If so, how to avoid the influence of these enzymes to the following viral RNA extraction?
4. suggest to add results of adenovirus assembly, e.g. the number and length of adenovirus contigs, to the results part.
5. why the phylogenetic tree of IVa2 is absent?
Reviewer 2 Report
Karamendin et al. reported their viromic analysis of Kazakhstani bats followed by phylogenetic characterization of a new adenovirus. Frankly, I was very surprised when seeing the title, because there are few data about bat-borne viruses in Central Asian countries.
The authors are encouraged to detect the viruses by a conventional method and to obtain the complete genome by PCR amplification. Besides, the author should introduce the lengths of the three sequences.
Please describe clearly how the authors treated these samples. Did they prepare the HTS library individually or collectively?
I doubt the robustness of phylogenetic analyses. In Fig. 1A, BtAdV-KZ01 is the target of phylogenetic analysis and shouldn’t be placed as an outgroup.
Lines 24-25: This sentence is a little bit overstated; please tone it down.
Line 26-28: The reference is too old, and the number 130 is out of date. Please visit the ZOVER database. Besides, I don’t agree that these viruses are pathogenic for humans.
Line 34: The present references don’t support that AdVs infect all vertebrates.
Line 50: Please change RPM to g.
Lines 53-55: The authors extracted RNA, but how did they construct the HTS library using the DNA kit?
Lines 78-80: Repetitive description.
Reviewer 3 Report
In this communication manuscript, the authors describe the detection of an adenovirus in bat samples collected in 2008 in South Kazakhstan. By metagenomics, the authors identified fragments related to bat mastadenovirus and suggested that it may belong to a new taxon based on sequence divergence and phylogenetic analysis.
The manuscript contains new data, as no bat adenoviruses have been reported from the country and there is limited knowledge about the diversity of adenoviruses in Kazakhstan. The methods are adequate, but I believe that the results need to be improved. In addition, there are some parts of the manuscript that authors must clarify, and additional data is requested to improve the quality of the manuscript, as indicated below.
Major points:
1. Line 77: Were all these reads assembled into just the 3 genes? How many reads were used to assemble each gene? The remaining reads, do they align with other viruses or with the host?
2. Table 1 indicates the percentage of similarity between the sequences, not exactly the evolutionary distance, which is defined as the number of substitutions per site that separates a pair of homologous sequences since they diverged from their common ancestral sequence. Authors should correct this in the manuscript or table.
3. How did you root the trees? It is really strange that for the hexon protein the new adenovirus is placed as an outgroup. I suggest authors to perform new phylogenetic analyzes using different evolutionary models (the best option is to use the model indicated by ModelTest from MEGA or IQtree) and with 1,000 bootstrap replicates. This could improve tree resolution, given the low bootstraps observed in many branches. Also, include in the methodology or figure legend the rationale for excluding some bootstrap values;
4. How can authors be sure that the gene sequences they found belong to the same virus? Since you haven't assembled the entire genome, additional analysis can be done to gain more confidence that the sequences actually belong to a unique viral species. I suggest you check the codon usage, %GC and dinucleotide patterns of the genes. If there are similar results, you can infer that the fragments belong to the same genome and your conclusions will be better supported;
5. Lines 109-111: I agree with the authors that the sequences belong to a putative new virus, but they should discuss this based on the criteria adopted by the ICTV Adenoviridae Study Group.
Minor points:
1. Line 47: italicize Myotis blythii;
2. Line 50: indicate the centrifuge rotor or indicate rotation as G;
3. Why the phylogenetic analyzes were performed with only two of the three genes?
4. In the new trees, make sure you include all GenBank accession numbers or include the numbers in a separate file as supplementary information;
5: Line 116: Italicize Pteropus dasymallus yayeyamae